# Miniature Ultralight Deformable Squama Mechanics and Skin Based on Piezoelectric Actuation

**DOI:** 10.3390/mi12080969

**Published:** 2021-08-16

**Authors:** Xiang Lu, Xiang Xi, Kun Lu, Chengxiang Wang, Xiang Chen, Yulie Wu, Xuezhong Wu, Dingbang Xiao

**Affiliations:** College of Intelligent Science, National University of Defense Technology, Changsha 410073, China; luxiang@nudt.edu.cn (X.L.); cx.wangnudt@nudt.edu.cn (C.W.); 19894369578@163.com (X.C.); ylwu@nudt.edu.cn (Y.W.); xzwu@nudt.edu.cn (X.W.); dingbangxiao@nudt.edu.cn (D.X.)

**Keywords:** squama, piezoelectric actuate, array, three-dimensional technology, skin

## Abstract

A miniature deformable squama mechanics based on piezoelectric actuation inspired by the deformable squama is proposed in this paper. The overall size of the mechanics is 16 mm × 6 mm × 6 mm, the weight is only 140 mg, the deflection angle range of the mechanical deformation is −15°~45°, and the mechanical deformation is controllable. The small-batch array processing of the miniature deformable squama mechanics, based on the stereoscopic process, laid the technological foundation for applying the deformed squama array arrangement. We also designed and manufactured a small actuation control boost circuit and a mobile phone piezoelectric control assistant application that makes it convenient to perform short-range non-contact control of the deformation of the squama. The proposed system arranges the deformed squamae into groups to form the skin and controlls the size and direction of the signals input to each group of the squama array, thereby making the skin able to produce different shapes to create deformable skin.

## 1. Introduction

The outer skin structure of some organisms is composed of squamae, which are the best protective layers for such organisms and can help them resist attack by predators and perform specific functions. For example, fish squamae reduce friction between the fish body and water and thus reduce the resistance of the fish during movement [1,2]. Studies have shown that the squamae of some organisms can be deformed adaptively in response to changes in the external environment. Inspired by biological squamae, some research institutions have designed and manufactured bionic squamae for industrial production [3,4,5,6]. Such artificial squamae resemble natural squamae in that they are lightweight and have good mechanical properties. Deformable squamae have broad application prospects, but there has been relatively little research on the array arrangement and adaptive deformation of squamae. Being able to change its deformed state allows the squama to take various forms and thereby deceive the outside world. It is also possible to arrange multiple deformed squamae in an array to form a deformable skin structure layer and produce different deformation effects by adjusting the deformed form of each squama.

In this paper, we propose a miniature deformable squama mechanics based on piezoelectric actuation. First, we introduce the principle of bionics and the design of the deformable mechanics. Next, we discuss the processing and manufacturing methods of deformed squamae and the design and manufacture of miniature actuation circuits. Finally, we describe the actuation and performance testing of the deformed squamae.

## 2. Materials and Methods

### 2.1. Design of Deformation Mechanics and Actuator

The various movements of natural organisms were mainly completed through the regulation of complex muscle movements with multiple degrees of freedom, as are the deformation movements of squamae. The artificial deformation mechanics designed in this study achieves complex movement by muscle actuation regulation, which will significantly increase a system’s weight, power, control, and manufacturing complexity. Studies have found that many organisms usually move their moving parts quickly and powerfully along a specific axis. Therefore, a simplified single-degree-of-freedom motion mechanics was used to achieve squama deformation—the form of single-degree-of-freedom deformation of the squama deflection along an axis. The specific manner of deflection is determined mainly by the design of the deformation mechanics and the actuation mechanics.

The overall size of the deformed squama mechanics is on the centimeter scale, and the characteristic size of the transmission mechanism is on the micrometer scale. The motion mechanics at the microscale needs to fully consider friction and other surface forces that hinder movement. Inspired by the transmission mechanics of the Robobee [7,8] developed by the team of Professor R J Wood of Harvard University, a spherical four-bar linkage mechanics was used as the scale deformation mechanics. A hinge mechanics can convert tiny deflections into visible angle changes.

A diagram of the structure of the connecting rod is shown in Figure 1a. Assuming that the clockwise rotation angle of the connecting rod mechanics is positive and that the input *δ* is downward, the following geometric relationships can be obtained:(1)$$L_{3}\cos\theta_{3}-L_{1}\sin\theta_{1}=L_{3}$$
(2)$$L_{1}\cos\theta_{1}+L_{3}\sin\theta_{3}-\delta =L_{1}$$
(3)$$\theta_{1}+\theta_{2}=\theta_{3}$$

The change in the output angle of the mechanics $\theta_{w}$ is equal to the angle $\theta_{3}$, and the above three geometric relation equations are combined to obtain the following results:(4)$$\theta_{1}=\sin^{-1}\left(\frac{L_{3}(\cos\theta_{3}-1)}{L_{1}}\right)$$
(5)$$\theta_{2}=\theta_{3}-\sin^{-1}\left(\frac{L_{3}(\cos\theta_{3}-1)}{L_{1}}\right)$$
(6)$$\delta =\sqrt{L_{1}^{2}-L_{3}^{2}{\left(\cos\theta_{3}-1\right)}^{2}}-L_{1}+L_{3}\sin\theta_{3}$$
where *δ* is the input, which is the deflection amplitude of the drive; $\theta_{w}$ is the output, which is the rotation angle of the scales, and $L_{2}=L_{4}$.

Macroscale actuation methods include the use of a motor drive, hydraulic actuation, and pneumatic actuation. However, because of the influence of the scale effect, the applicability of these actuation methods at the microscale is quite limited. Among the more applicable actuation methods at the microscale are magnetic actuation [9,10,11,12], electrostatic actuation [13], piezoelectric actuation [14,15,16,17,18], photothermal actuation [19,20], acoustic actuation [21,22], chemical actuation [23,24], and biological actuation [25,26]. Biological actuation is an ideal actuation method, but it has only been developed in theoretical research. Piezoelectric actuation is one of the most widely studied drive methods. Its main principle is the generation of power by means of control of the deformation of a material. This can result in strong actuation force and displacement characteristics, rapid response, a simple structure, high energy utilization, and wide application potential in micro/nano robots. However, the voltage required for piezoelectric actuation is relatively large; therefore, the design requirements of the actuation circuit are higher. Electrostatic actuation and acoustic actuation are characterized by a simple structure and enhanced performance as the degree of miniaturization increases, and therefore, they have great application prospects in the field of nano-robots. Magnetic actuation is characterized by a large actuation force and a simple structure, but it needs to be controlled by an external magnetic field. Photothermal actuation is a non-contact power drive method with a fast response and no electromagnetic interference. However, the actuation principle and manufacturing process are more complex than the previously mentioned methods. Chemical actuation mainly uses chemical reactions to drive energy, which results in a high energy utilization rate and low cost. This method is limited, however, by insufficient reaction process control accuracy and short action times. Based on a comprehensive comparison of their characteristics, piezoelectric actuation was judged to be the best choice for use in this study, in terms of deflection, size, and accuracy.

Research has confirmed that when the planar shape of a piezoelectric actuator is almost triangular (an equal-strength beam), the utilization rate of the material, the bending resistance of the overall structure, and the mechanical energy density can reach their highest levels [15]. The plane profile of the piezoelectric actuator considered in this study is shown in Figure 1b. The length of the piezoelectric ceramic sheet is l; the widths of the head and tail parts are $w_{1}$ and $w_{2}$, respectively; the length of the extended end is $l_{1}$. Based on the design of the equal-strength beam, $l_{1}$ can be determined using the following formula:(7)$$l_{1}=\frac{w_{2}}{w_{1}-w_{2}}l$$

The dimensions of the final deformation mechanics and driver are shown in Table 1.

The actuator and the transmission mechanics were integrated to obtain the deformed squama mechanics. A view of the deformed mechanics in three directions is shown in Figure 2a. The *L*_3_ rod at the connection between the mechanics and the scale had an I-shaped structure, which was convenient for positioning and could increase the contact area between the scale and the deformation mechanics and thereby improve the stability of the structural connection. The insulating part at the end of the piezoelectric actuator was fixed to the slot at the base of the deformation mechanics. The insulating portion at the head end was inserted into the slot of the spherical four-bar transmission mechanics. When the input signal causes the drive mechanics to deflect, the change at its head end is the largest, and the spherical four-bar transmission mechanics amplifies it into the deflection of the scales connected to the *L*_3_ rod.

The deformation method achieved using this experimental mechanics is a rotation of the scales along the design axis *L*_3_ at any fixed angle in the range of −15°~45°, as shown in Figure 2c. A deflection angle in the counterclockwise direction is taken to be positive. The maximum deflection angle in the clockwise direction is only 15° because the height of the mechanics and the area of the scales spatially limit the amount of deflection possible. The size of the mechanics can be adjusted to change the clockwise angle threshold.

### 2.2. Processing and Manufacturing of Deformation Mechanics and Actuator

The deformation mechanics was made of rigid–flexible composite materials. A three-dimensional model structure of the mechanics is shown in Figure 2. The overall size was 16 mm × 6 mm × 6 mm. The rotating hinge in the spherical four-bar linkage transmission mechanics was replaced by a flexible film. When rotating, only the film is deformed; the rigid connecting rods are not in contact with each other, and therefore, the huge energy loss that would be caused by rotating friction on a small scale is avoided, and the connecting rod structure can move flexibly at the microscale.

The deformation mechanics was processed by an integrated molding process based on the stereoscopic process [27,28]. The rigid layer was made of a strong and light unidirectional carbon fiber sheet with a thickness of 70 µm; the flexible layer was made of high-temperature-resistant and non-deformable polyimide film with a thickness of 12.5 µm; the adhesive layer was made of a protective waterproof high-viscosity polymer resin layer with a thickness of 15 µm. The processing process, illustrated in Figure 3a, consists of the following steps:The carbon fiberboard and resin glue were cured and thermal stress was released so that the carbon fiber board layer remained flat after gluing;Laser etching was used to etch the designed hinge groove on the carbon fiberboard and the cured resin glue layer;The polyimide film layer was cured in alignment with the adhesive layer between the two pretreated carbon fiber boards and the resin adhesive curing layer;A stress release treatment was performed to keep the rigid–flexible composite material layer flat;The outer contour of the cured rigid–flexible composite material layer was released, and the released flat rigid–flexible composite material layer was folded according to the designed structure’s fixed-body deformation mechanics.

The processing of this deformation mechanics was performed using a small-batch array processing method. Overall, 10 deformation mechanicss could be processed at a time within the range of a 100 mm × 100 mm plate, as shown in Figure 3b–d.

The piezoelectric actuator mainly comprised two layers of 120 µm thick PbZrTiO3(PZT)-5H piezoelectric ceramics, two layers of 120 µm thick glass fiber, and a layer of 30 µm thick carbon fiber prepreg. The structure is shown in Figure 4a. The total length was approximately 15 mm. The curing model is shown in Figure 4b. First, the customized 20 mm × 20 mm PZT-5H piezoelectric sheet, glass fiber, and prepreg were processed into the shape required by drawing and by laser cutting. They were then stacked in the order of design, put into a customized curing process fixture, and cured for 90 min at 120 °C and 1 atm, so that the PZT-5H piezoelectric ceramic sheet and glass fiber firmly adhered to the carbon fiber intermediate layer. They were then released by laser etching. Finally, lead wires were attached to the piezoelectric actuator. Each actuator needed three wires leading out, one each to the upper and lower piezoelectric sheet and one to the carbon fiber layer in the middle. The lead wires were attached by soldering or conductive silver glue bonding. As shown in Figure 4c, the piezoelectric actuator in this experiment also used small-batch array processing, which could be processed at one time on a 50 mm × 50 mm processing range plate of 16 piezoelectric actuators. This improves the processing efficiency, reduces the alignment operations of each processing structure, and improves the consistency of the actuators.

### 2.3. Miniature Boost Actuate Circuit and Wireless Control Software

The driving voltage of the piezoelectric driver needs to reach hundreds of volts. To generate the high-voltage signals required to drive the piezoelectric actuator, we designed and fabricated a custom lightweight boost converter circuit that can convert a low-voltage (12 V) direct-current (DC) input signal into a signal in the range of 12 to 250 V. The main components of the circuit, shown in Figure 5, are a single-chip microcomputer, switch interfacing circuit (IC), field-effect tube, inductor, operational amplifier, and diode. The circuit was designed to increase the voltage of the low-voltage DC battery in a step-less fashion using the boost architecture principle. To facilitate the operation of the circuit control, a Bluetooth module was added to the circuit, and a mobile phone piezoelectric Bluetooth assistant application was designed and developed. In this application, the Bluetooth module of the circuit can be connected through the mobile phone’s Bluetooth to transmit the external signal output to the circuit. Real-time control makes it easy to adjust the input voltage of the driver as required at any time. The input current of the low-voltage power supply corresponding to the specified output signal is shown in Figure 6. It can be seen that the current and power consumption of the circuit increase approximately linearly.

## 3. Results and Discussion

### 3.1. Single Squama Deflection Angle Test

In this study, the processed piezoelectric actuator and the main body of the deformation mechanics were assembled and fixed according to the design method. The piezoelectric actuator deformation mechanics obtained is shown in Figure 7. Given a specific signal to the actuator, the I-shaped rod on the deformation mechanics can be deflected, the squamae adhere to the I-shaped rod, and the center of the squamae coincides with the center of the I-shaped connecting rod to keep the initial state stable. The squama used in this experiment was a carbon fiberboard with a special wave-absorbing material sprayed on the surface. The size of a single squama was 300 mm × 300 mm, and the mass was 0.5 g, and the maximum load was 1.2 g.

The driver was a piezoelectric bi-crystal ceramic driver that makes the I-shaped rod deflect in two directions. As shown in Figure 8, when the given driving signal is a forward-biased voltage, the squamae follow the I-shaped link to deflect counterclockwise, and the angle of deflection gradually increases as the driving voltage increases. When the driving voltage reaches the maximum of 250 V, the deflection angle of the squama reaches the design threshold of 45°. Table 2 lists the actual scale angles corresponding to given forward-bias driving voltages.

When the given driving signal is reverse biased, the squamae follow the I-shaped link to deflect clockwise, as shown in Figure 9. Due to the height limitation of the mechanics, when the given driving signal reaches 180 V, the squamae are deflected to the base and reach the −15° limit of clockwise deflection. Table 3 lists the squama angles corresponding to given reverse-biased drive voltages. Figure 10 shows the output bias angles corresponding to the forward-biased and reverse-biased drive signals. The output biases obtained for the same bias signal magnitude in the two directions are inconsistent because of the asymmetry of the force in the two directions of the drive mechanics and machining errors. Additionally, there are currently two measures to reduce this error: The first measure is to reduce the original processing error by using higher-precision laser etching equipment; the other is to improve the accuracy of manual folding by designing an auxiliary folding mechanism.

### 3.2. Arrange Deformed Squamae in Array to Form a Deformable Skin

A single squama is small in size; thus, it is rarely used alone. Several deformed squamae are usually arranged according to a particular design to form a deformed squama group. The deformation of each squama in the deformed scale group can be controlled individually or in groups. In this experiment, a simple 2 × 3 rectangular deformed squama array was used for the verification test. In this squama array, one input signal could be connected to six squamae at the same time, so that the same angle could deflect the six squamae at the same time according to one control, or the input signal could be divided into two or three independent channels to make every two or three squamae in the array form a small group. The deformation within the group was the same, and the different groups did not interfere with each other, deforming independently to form different deformation combinations. As shown in Figure 11, the squamae were divided into three groups, every two squamae forming a group, and each group was given an independent drive signal to control the deflection of the squamae.

Figure 12a shows the state of the squama array with no input signal. Figure 12b shows th deformation mode when the drive signals given to the three groups were all forward biased at 150 V. Figure 12c shows the deformation combination when the driving signals provided to the three groups were a 100 V forward biased, a 150 V forward biased, and a 200 V forward biased. Figure 12d shows the deformation combination when the driving signals given to the three groups were a 150V forward biased, a 150 V reverse biased, and a 150 V forward biased.

The designed array arrangement of the deformed squamae can form the skin of specific structures and devices. Controlling the size and direction of the signal input to each group of the squama array makes it possible to form the deformed squama array into many different deformation combinations, thereby making the skin produce different shapes to create deformable skin.

## 4. Conclusions

This paper proposes a miniature deformable squama mechanics based on a piezoelectric actuator whose overall size is 16 mm × 6 mm × 6 mm and the mass is only 140 mg. The deflection angle range of the mechanical deformation is −15°~45°, and the drive signal can be controlled to fix the squamae at any angle within this range. The processing of the squama deformation mechanics relies on a small batch array processing method based on the stereoscopic process that can achieve ten deformation mechanics through one processing sequence of the material in the range of 100 mm × 100 mm. This method has high processing efficiency and ensures that the product has a good consistency. A small boost converter circuit was designed and manufactured to convert the input 12 V low-voltage direct-current signal into any signal in the range of 12 to 250 V and connect a mobile phone application via Bluetooth to control the change of the signal. Arranging the deformed squamae into groups to form the skin and controlling the size and direction of the signal input to each group of the squama array makes the skin produce different shapes of deformable skin.

However, more in-depth and systematic research is needed in the future, and the scope and consistency of the mechanical deformation need to be further optimized to improve the driving accuracy of the mechanics, the driving force, the power supply of the system, and the ultra-miniature embedding of the circuit system.

## Figures and Tables

**Figure 1 micromachines-12-00969-f001:**
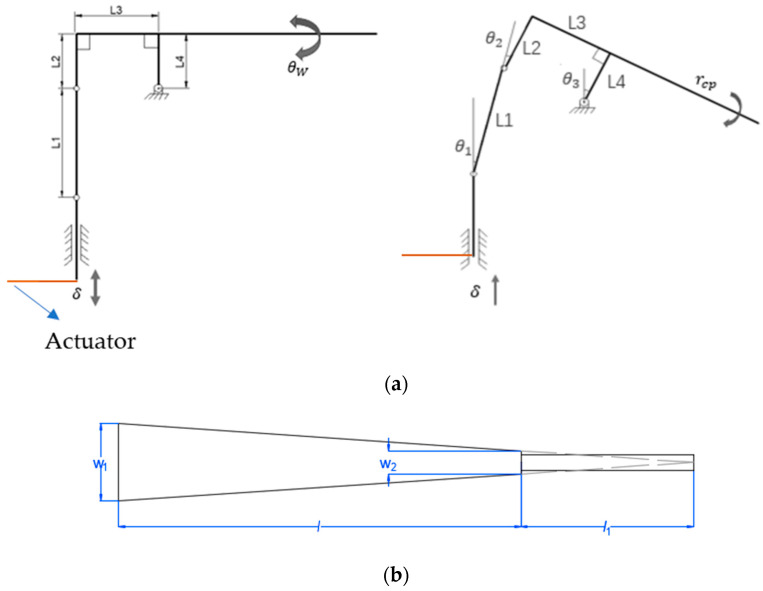
Design of deformation mechanics and actuator: (**a**) diagram of the structure of the spherical four-bar linkage; (**b**) plane profile of the piezoelectric actuator.

**Figure 2 micromachines-12-00969-f002:**
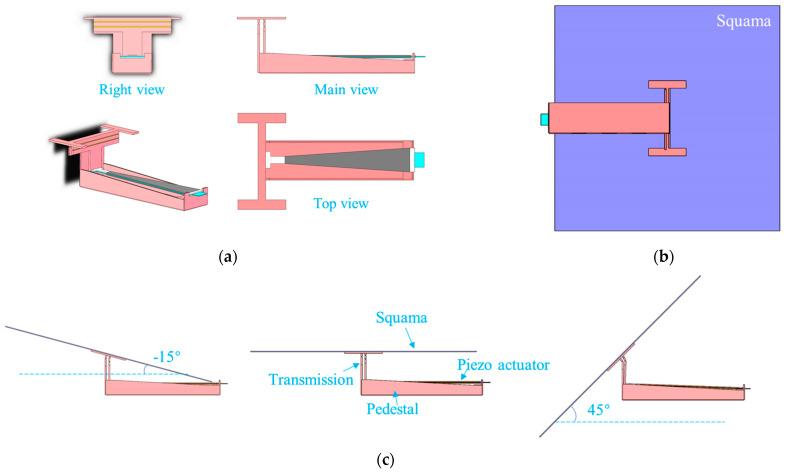
Deformable squama mechanics: (**a**) view of the deformed mechanics in three directions; (**b**) deformable mechanics model with squama installed; (**c**) deflection range of deformed squama, −15°–45°.

**Figure 3 micromachines-12-00969-f003:**
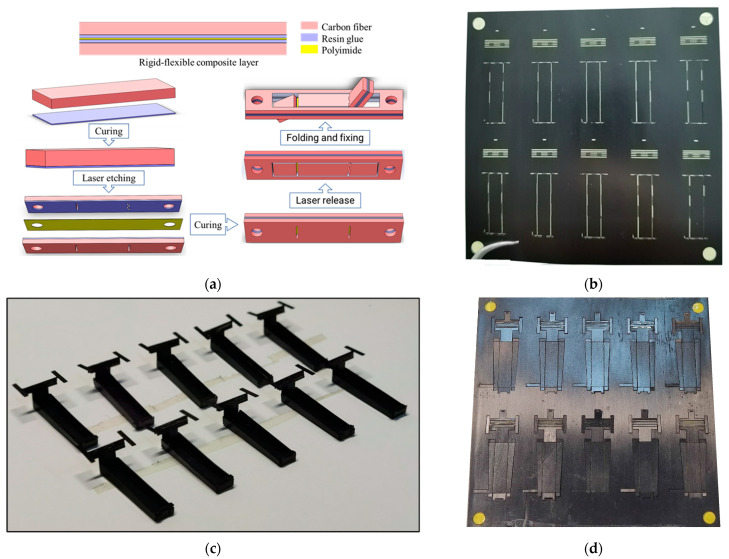
Manufacturing of deformation mechanics: (**a**) processing flow of deformation mechanics; (**b**) array processing of deformation mechanics’s hinge holes; (**c**) batch release of deformation mechanics; (**d**) deformation mechanics processed at one time.

**Figure 4 micromachines-12-00969-f004:**
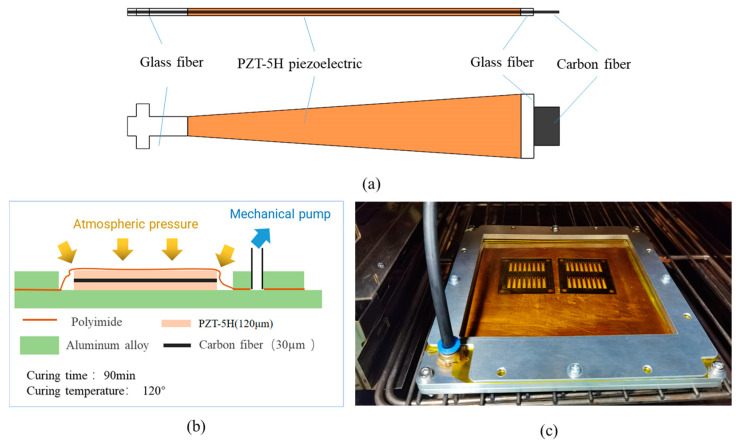
Manufacturing of piezoelectric actuator: (**a**) structure of the piezoelectric actuator; (**b**) curing model of the piezoelectric actuator; (**c**) arrayed manufacturing of actuator.

**Figure 5 micromachines-12-00969-f005:**
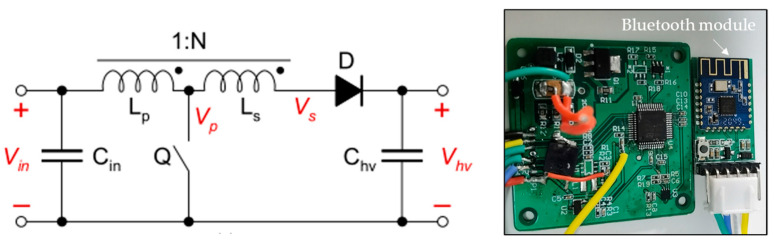
Miniature boost actuation circuit.

**Figure 6 micromachines-12-00969-f006:**
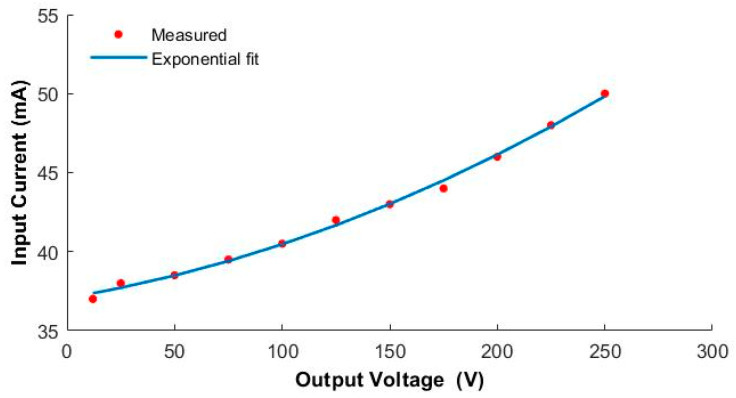
Low-voltage input current versus high-voltage output generated by the boost converter.

**Figure 7 micromachines-12-00969-f007:**
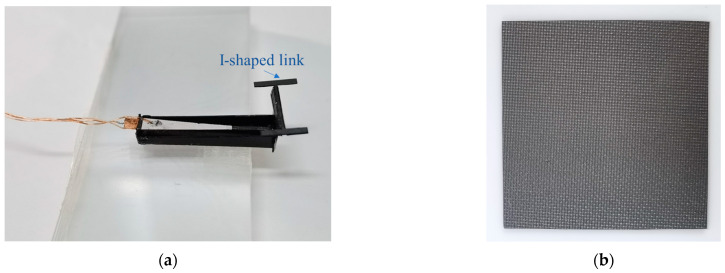
(**a**) Experimental product of miniature piezoelectric drive squama deformation mechanics; (**b**) squama.

**Figure 8 micromachines-12-00969-f008:**
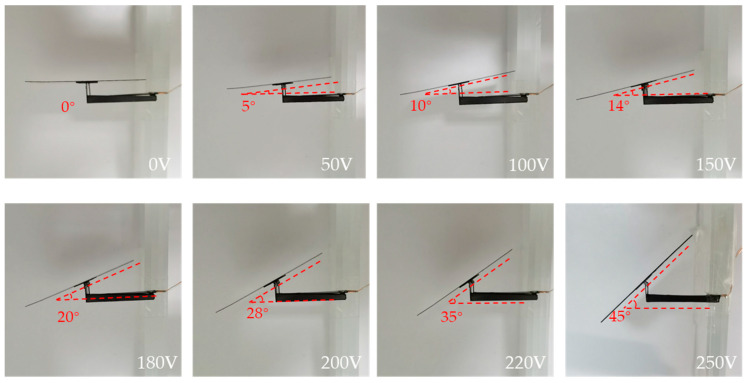
Forward-bias drive test of deformed squama.

**Figure 9 micromachines-12-00969-f009:**
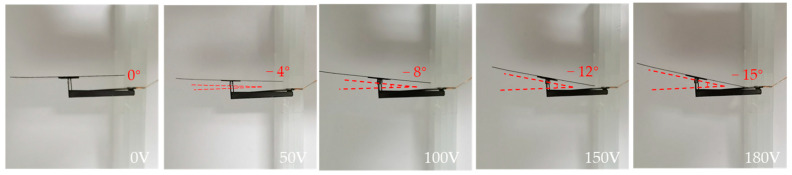
Reverse-biased drive test of deformed squama.

**Figure 10 micromachines-12-00969-f010:**
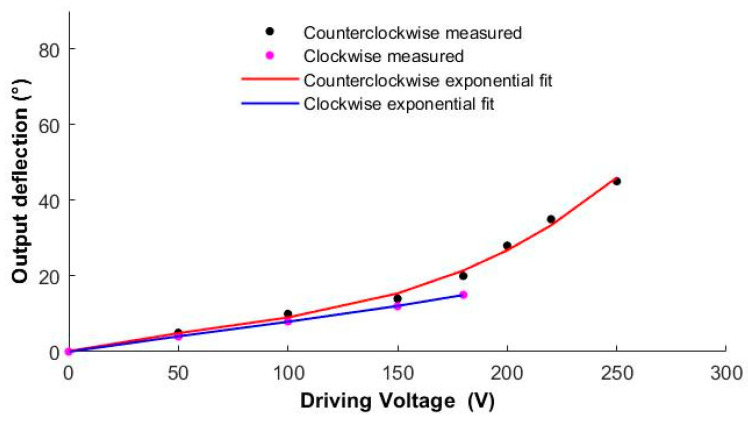
Output bias angle versus forward-biased and reverse-biased drive signals.

**Figure 11 micromachines-12-00969-f011:**
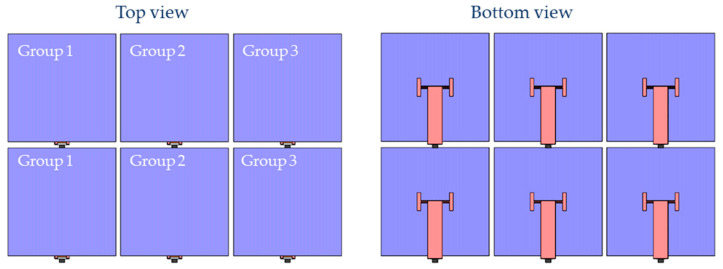
Arrange deformed squamae in an array to form deformable skin.

**Figure 12 micromachines-12-00969-f012:**
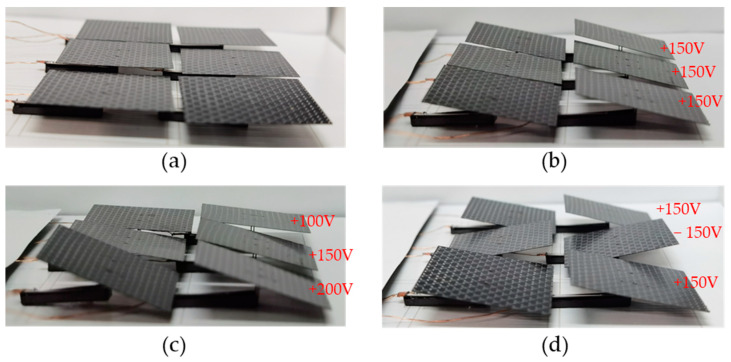
Tests of deformable skin: (**a**) no signal input; (**b**) drive signals given to the three groups all forward biased at 150 V; (**c**) drive signals given to the three groups are 100 V forward biased, 150 V forward biased, and 200 V forward biased; (**d**) drive signals given to the three groups are 150 V forward biased, 150 V reverse biased, and 150 V forward biased.

**Table 1 micromachines-12-00969-t001:** The specific dimensions of the final deformation mechanics and driver.

**Structural Part**	$L_{1}$	$L_{2}$	$L_{3}$	$L_{4}$	l	$w_{1}$	$w_{2}$	$l_{1}$
**Size (mm)**	0.35	0.3	0.3	0.5	13	2.5	0.75	5.5

**Table 2 micromachines-12-00969-t002:** Actual scale angles corresponding to given forward-bias driving voltages.

**Driving Voltage (V)**	0	50	100	150	180	200	220	250
**Output Deflection (°)**	0	5	10	14	20	28	35	45

**Table 3 micromachines-12-00969-t003:** The actual scale angle corresponding to the given reverse-biased driving voltage.

**Driving Voltage (V)**	0	50	100	150	180
**Output Deflection (°)**	0	−4	−8	−12	−15

## Data Availability

Data are contained within the article.

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
