# Peer review of "Miniature Ultralight Deformable Squama Mechanics and Skin Based on Piezoelectric Actuation"

_micromachines, 2021, doi:10.3390/mi12080969_

Round 1

Reviewer 1 Report

Authors have proposed a  miniature deformable squama mechanism and designed different mechanism in order to control the process. It is an original study, which has been successfully described and developed. Congratulations to authors 

Author Response

Thank you for your valuable suggestions and positive comments. Your opinion is an affirmation of me as well as the essential guiding significance to me, and I will work harder to make better results.

Reviewer 2 Report

The paper decribes a novel approach to elaborate light weight, electronically controlled scale arrangements based on piezoelectric actuated movements.

It is a nice, interesting description of a prototype with indication of further improvements.

The following remarks and questions could be addressed:

- The word mechanism might be a bit misleading at certain areas, please consider the use of the word "mechanics" instead of mechanism".

- What is the possible maximum load, i.e. the maximum possible weight of scale (squama) to be used in the system ?

- What could be the improvement to overcome the output bias shown on Fig 10?

Some formal remarks:

- abstract, 13th row: Is it -15-45o or 15-45o ?

- hyphens between numbers and units (e.g. 100-V in Figure 2. caption) appear, it should be removed

- 59th row: „characteristic size is on the micrometer scale” – should be more clear, the characteristic size of which feature?

-127-132 row paragraph – it seems inconsistent what is minus value and clockwise, please check

- Fig 7: please indicate a) and b) with figure caption

- I would suggest to name the „Results” section as „Results and Discussion” as basically the analysis is shown in this section as well

- „Conclusion” word spelling
